# A Patch-based Student-Teacher Pyramid Matching Approach to Anomaly Detection in 3D Magnetic Resonance Imaging

**Johannes Schwarz** [1,4]                    JOHANNES.SCHWARZ@RUHR-UNI-BOCHUM.DE

**Lena Will** [3,4]                    LENA.WILL@KK-BOCHUM.DE

**Jörg Wellmer** [4]                    JOERG.WELLMER@KK-BOCHUM.DE

**Axel Mosig** [1,2]                    AXEL.MOSIG@RUHR-UNI-BOCHUM.DE

[1] *Center for Protein Diagnostics, Ruhr University Bochum, Germany*

[2] *Faculty of Biology and Biotechnology, Bioinformatics Group, Ruhr University Bochum, Germany*

[3] *Department of Diagnostic and Interventional Radiology, Neuroradiology and Nuclear Medicine, University Hospital Knappschaftskrankenhaus Bochum, Ruhr-University Bochum, Germany*

[4] *Ruhr-Epileptology, Department of Neurology, University Hospital Knappschaftskrankenhaus Bochum, Ruhr-University Bochum, Germany*

**Editors:** Accepted for publication at MIDL 2024

## Abstract

Anomaly detection on 3D magnetic resonance images (MRI) is of high medical relevance in the context of detecting lesions associated with different diseases. Yet, reliable anomaly detection in MRI images involves major challenges, specifically taking into account information in 3D, and the need to localize relatively small and subtle abnormalities within the context of whole organ MRIs. In this paper, a top-down approach, which uses student-teacher feature pyramid matching (STFPM) for detecting anomalies at image and voxel level, is applied to 3D brain MRI inputs. The combination of a 3D patch based self-supervised pre-training and axial-coronal-sagittal (ACS) convolutions pushes the performance above that of f-AnoGAN (bottom-up). The evaluation is based on a tumor dataset. Our code is available on GitHub (3D-STFPM-3DSSPL-ACS).

**Keywords:** magnetic resonance imaging, anomaly detection, semi-supervised learning, student-teacher feature pyramid matching, voxel and image-level detection

## 1. Introduction

In medical image analysis, it is often an attractive and promising approach to view the identification and localization of disease as an anomaly detection problem, i.e., to regard the recognition of disease patterns as the identification of deviations from the norm. While unsupervised or semi-supervised anomaly detection approaches limit the need for extensive annotation, its application in specific medical settings is hampered by several factors. Most current anomaly detection approaches were devised for 2D images and thus do not accommodate the 3D nature of magnetic resonance imaging (MRI). Furthermore, the predominant bottom-up or generative approaches only work well on smaller scaled volumes (Simarro et al., 2020).

We here investigate a novel approach to anomaly detection based on the Student-Teacher model (Bergmann et al., 2019; Wang et al., 2021). This approach is based on coupling the

training process of two convolutional networks: A teacher network, which is trained on normal images as well as images containing anomalies, and a student network, which is trained on normal, anomaly-free images only. The basic idea of a student-teacher network is that when presented with an image containing anomalies, the image representation of the teacher will deviate significantly from the image representation of the student model. An anomaly map is formed by the difference between the layers of the teacher and the student. The conceptual advantage over generative models for detecting anomalies is that the student-teacher is a purely top-down approach and thus allows an explicit definition of a classification loss. As a novel contribution to student-teacher networks, our approach presented here employs triplet margin loss (Vassileios Balntas and Mikolajczyk, 2016) for 3D self-supervised patch learning.

A further limiting factor for utilizing the student-teacher approach in medical applications is the reliance on ImageNet-pretrained networks, so that the teacher network cannot be trained with domain-specific knowledge. To address this, our 3D student-teacher feature pyramid matching (STFPM) network uses the top-down structure of the ResNet to connect it to self-supervised patch learning. Here, this technique is applied to the teacher network in the form of patch learning to be able to provide the teacher with a wide range of input data (Danon et al., 2018).

Although not required in our approach, pre-training is still useful to deal with notoriously limited training data in medical applications. To facilitate pre-training in our 3D MRI setting, we employ Axial-Coronal-Sagittal (ACS) convolutions (Yang et al., 2019), which allow the use of 2D ImageNet weights for 3D convolutions.

We evaluate our approach using a data set combining the BraTS tumor data set (Baid and Ghodasara, 2021; Menze et al., 2015; Bakas et al., 2017) with healthy MRI data from the IXI data set (IXI). We show that the use of ImageNet weights and self-supervised patch learning has a major impact on the performance of the 3D STFPM.

This combination of the student-teacher approach with patch-based learning and ACS convolutions creates a network that can detect anomalies both at the image level and at the pixel level.

To summarize, the main contributions of the paper are: **1.** the first approach that extends STFPM to 3D MRI scans; **2.** the use of self-supervised patch learning (SSPL) on 3D MRI scans; and **3.** the combination of ACS-Convolution and ImageNet pre-trained weights on a medical domain.

## 2. Related Work

Anomaly detection is often performed via generative networks. This includes GAN-based (Schlegl et al., 2017, 2019; Akcay et al., 2018; Donahue et al., 2016) approaches as well as autoencoders (An and Cho, 2015). In medical imaging on 3D MRI scans, a low resolution is usually selected for GAN-based networks (Schlegl et al., 2019; Siddiquee et al., 2019; Luo et al., 2023) and for autoencoders (Behrendt et al., 2022; Pinaya et al., 2021; Baur et al., 2021). Slices are often also extracted from the axial, sagittal or coronal and anomaly detection is only operated at slice level (Pinaya et al., 2021; Han et al., 2020).

The student-teacher approach was applied to the MVTec data set and has proven its effectiveness here (Bergmann et al., 2019; Wang et al., 2021). Self-supervised patch learning

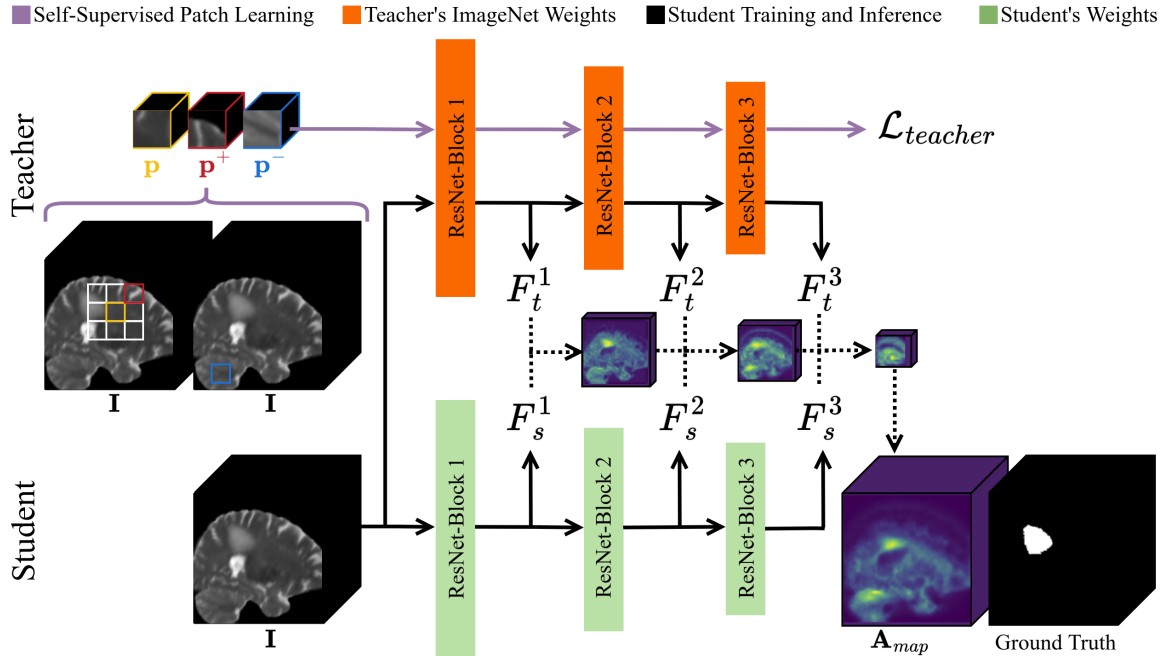

Figure 1: Schematic structure of the 3D student teacher pyramid matching framework, with 3D self-supervised patch learning ($\mathbf{p}$, $\mathbf{p}^+$, $\mathbf{p}^-$). In contrast to the teacher, the student only knows how healthy tissue is represented in its interlayers. By comparing the intermediate layer outputs from the teacher $F_t$ and student $F_s$, a 3D anomaly map $\mathbf{A}_{map}$ is generated.

comes from the field of metric learning and describes how local image descriptors can be learned. In connection with the student-teacher approach, self-supervised patch learning was already mentioned in Bergmann et al. (2019). Here, however, it was not applied to medical images and only to 2D data.

## 3. Method

In this work, the Student Teacher Model, introduced by Wang et al. (2021), is used and extended to implicitly learn the feature distribution of healthy and diseased data. Both networks, student and teacher, are based on the same architecture to minimize knowledge loss (Hinton et al., 2015).

The teacher network is pre-trained on healthy and diseased MRI's using self-supervised patch learning. On the other hand, there is the student, which only sees data from healthy patients and is not pre-trained. Pre-training the teacher is done through the following steps: Use Axial-Coronal-Sagittal (ACS) convolution (Yang et al., 2019) to make use of ImageNet weights in 3D convolution and self-supervised image patch learning (Danon et al., 2018). Then, patch learning allows the teacher to recreate healthy and non-healthy tissue in the

feature maps. After the pre-training is complete, the teacher's weights are frozen and only the student's weights are trained. Both the teacher and the student receive an input image, $\mathbf{I} \in \mathbb{R}^{C \times X \times Y \times Z}$, where $C$ is the channel dimension, and $X$, $Y$ and $Z$ represents the 3D image resolution, respectively. Then feature embeddings of the student and the teacher are computed after each ResNet block $R_{block}^i$ (with $i \in \{1, 2, 3\}$). In this way, the student is meant to learn how normal samples are distributed. During testing and evaluation, the difference between feature maps $F_t$ and $F_s$ are calculated, scaled up, and multiplied which each other and further used as anomaly map $\mathbf{A}_{map}$ with the meaning of the higher the difference, the higher the probability of an anomaly.

### 3.1. Pre-training the Teacher

A ResNet is used as the basis of the teacher. All convolutional layers are replaced by ACS convolutional layers, while the pre-trained ImageNet weights are retained.

**ACS convolution**   In ACS convolution (Yang et al., 2019), 2D convolutions are performed in three directions (axial ($a$), coronal ($c$), sagittal ($s$)) of the 3D volume. For this, the 2D kernel is split into three 3D kernels:

$$\mathbf{W}_a \in \mathbb{R}^{C_{in} \times C_{out}^{(a)} \times K \times K \times 1}, \mathbf{W}_c \in \mathbb{R}^{C_{in} \times C_{out}^{(c)} \times K \times 1 \times K}, \mathbf{W}_s \in \mathbb{R}^{C_{in} \times C_{out}^{(s)} \times 1 \times K \times K} \tag{1}$$

From a 3D input feature $\mathbf{I}_{in} \in \mathbb{R}^{C_{in} \times X \times Y \times Z}$, a 3D output $\mathbf{I}_{out} \in \mathbb{R}^{C_{out} \times X \times Y \times Z}$ is created with the ACS convolution, which uses the 2D convolutional kernel. $C_{in}$ and $C_{out}$ are the input and output channels, and $K$ denotes the kernel size.

**Self-Supervised Image Patch Learning**   After the ImageNet weights have been initialized, the teacher is trained using self-supervised patch learning which, following Bergmann et al. (2019) and Danon et al. (2018), yields local image descriptors as a result. In this work, we extended the approach from Danon et al. (2018) to 3D input images.

For this purpose, anchor boxes $\mathbf{p}$ of spatial size $(32 \times 32 \times 32)$ are randomly cut out for each image $\mathbf{I}$ in the teacher training. Then, following a grid, another box $\mathbf{p}^+$ is cut out in the immediate vicinity (positive box). Negative patches are cut out from another MRI image at a random position. Each of the boxes has a spatial size of $(32 \times 32 \times 32)$. Anchor boxes that only contain background voxels are discarded.

As Bergmann et al. (2019) suggests, in-triplet hard negative mining with anchor swap (Vassileios Balntas and Mikolajczyk, 2016) is used as a loss function that implements an embedding sensitive to the $\ell_2$ metric:

$$\mathcal{L}_{teacher} = \max \{0, \delta + \delta^+ - \delta^-\} \tag{2}$$

where $\delta > 0$ denotes the margin parameter and in-triplet distances $\delta^+$ and $\delta^-$ are defined as:

$$\delta^+ = ||\hat{T}(\mathbf{p}) - \hat{T}(\mathbf{p}^+)||^2 \tag{3}$$

$$\delta^- = \min\{||\hat{T}(\mathbf{p}) - \hat{T}(\mathbf{p}^-)||^2, ||\hat{T}(\mathbf{p}^+) - \hat{T}(\mathbf{p}^-)||^2\} \tag{4}$$

$\hat{T}$ is the output of the Teacher ResNet.

### 3.2. Training the Student

The student's training uses the same architecture as the teacher's to achieve optimal knowledge distillation. This means that all convolution layers are replaced by an ACS convolution. Only, there is no pre-training and the weights are initialized randomly.

**Training**  After the teacher has been pre-trained, their weights are frozen and only the student's weights are trained, which is initialized with randomized weights. In each student training step, a batch $\mathcal{B}$ of images $\mathcal{B} = \mathbf{I}_1, \mathbf{I}_2, \ldots, \mathbf{I}_n$ is fed into the teacher and into the student. Given an input image, $\mathbf{I} \in \mathbb{R}^{C \times X \times Y \times Z}$ the features $F_t^l(\mathbf{I})$ and $F_s^l(\mathbf{I})$ are calculated after each ResNet block $l \in \{1, 2, 3\}$. To calculate the loss at all positions $(x, y, z)$ in the feature maps, a $\ell_2$ distance between $\ell_2$ normalized feature vectors is defined and thus the loss over the whole image is calculated via the average at each image position:

$$\mathcal{L}_{student}^l(\mathbf{I}) = \frac{1}{X_l Y_l Z_l} \sum_{x=1}^{X_l} \sum_{y=1}^{Y_l} \sum_{z=1}^{Z_l} \left( \frac{1}{2} \left|\left| \hat{F}_t^l(\mathbf{I})_{xyz} - \hat{F}_s^l(\mathbf{I})_{xyz} \right|\right|_{\ell_2}^2 \right) \tag{5}$$

Here $X^l$, $Y^l$ and $Z^l$ are the spatial resolution of the feature map by ResNet block $l$. As in Wang et al. (2021), the features $F_t^l(\mathbf{I})$ and $F_s^l(\mathbf{I})$ are respectively normalized to form $\hat{F}_t^l(\mathbf{I}) = \left( F_t^l(\mathbf{I}) \right) / \left( \left|\left| F_t^l(\mathbf{I}) \right|\right|_{\ell_2}^2 \right)$ and $\hat{F}_s^l(\mathbf{I}) = \left( F_s^l(\mathbf{I}) \right) / \left( \left|\left| F_s^l(\mathbf{I}) \right|\right|_{\ell_2}^2 \right)$.

**Evaluation**  For an image $\mathbf{I}$ that is to be evaluated, the features of the teacher $F_t^l(\mathbf{I})$ and the student $F_s^l(\mathbf{I})$ are calculated and then scaled up by trilinear interpolation to the size of the input image $\mathbf{I}$. The upscaled images are each multiplied with one another, resulting in an anomaly map $\mathbf{A}_{map}$. To obtain a detection score $s_{detect}$ for an image $\mathbf{I}$, the maximum value of $\mathbf{A}_{map}$ is used: $s_{detect} = \max(\mathbf{A}_{map})$. The entire anomaly map $\mathbf{A}_{map}$ is used for pixel level detection.

## 4. Experiments

All experiments and their evaluation are performed on the BraTS 2021 (Baid and Ghodasara, 2021; Menze et al., 2015; Bakas et al., 2017) and IXI (IXI) data set. To find the optimal model and show that the combination of ACS convolution, patch learning and pre-trained ImageNet weights delivers state-of-the-art performance, the following experiments were performed: **Experiment 1**: The teacher is initialized with ImageNet with no further teacher training, constituting a fully unsupervised learning setting. **Experiment 2**: The teacher is not initialized with ImageNet weights, but the self-supervised patch learning is applied. **Experiment 3**: The teacher is initialized with ImageNet weights, and the self-supervised patch learning is used in addition. **Experiment 4**: Same as Experiment 3, but k-means clustering is applied in the evaluation of the $\mathbf{A}_{map}$ anomaly map, as suggested by Siddiquee et al. (2019). Assuming that there is always one contiguous lesion, two clusters can be formed. Namely, one cluster for the healthy tissue and one cluster for the diseased tissue. Since only lesional images are used for this, no detection performance is given here (see table 1).

For all experiments, ACS convolution is used for the teacher and the student. The experiments aiming to use tumors on the BraTS dataset each use the T2 sequence, since

the IXI (IXI) dataset only provides a T2 sequence and no FLAIR sequence. As a reference method for anomaly detection, we use the f-AnoGAN from Schlegl et al. (2019). The f-AnoGAN is an unsupervised method that detects anomalies both at the image level and at the pixel level. Originally, the f-AnoGAN can only process 2D images. For comparison, all 2D convolutional layers have been replaced by 3D convolutional layers, similar to Simarro et al. (2020).

### 4.1. Preprocessing

For comparability, all images are registered on the template MNI-152 (Manera et al., 2020) and skull-stripped with a prefabricated mask to avoid hyperintensities. In addition, histogram standardization (Nyul et al., 2000) and Z-normalization is performed since the MRIs come from different sources (healthy from IXI dataset, diseased from BraTS dataset). The MRI scans are cropped to $(156 \times 156 \times 156)$ after registration, and then scaled to $(224 \times 224 \times 224)$. This is done to eliminate the large black borders around the MRIs that appear just after skull stripping.

### 4.2. Dataset

BraTS and IXI data sets are each split into training (70%), validation (15%), and test (15%) data sets. For the training of the teacher, the same number of MRI scans are taken from the pool of training data of the BraTS and the IXI data set. All IXI images from the training pool are used for the training of the student. All BraTS images from the validation or test pool are used to evaluate the segmentation performance. Equal amounts of IXI and BraTS data from the validation and test pools are used to evaluate the classification performance. The segmentation map of the BraTS data set consists of several regions (no lesion - label 0; non-enhancing tumor core - label 1; the peritumoral edema - label 2; GD-enhancing tumor - label 4). To generate a binary segmentation map, all values greater than 0 are considered a lesion (Baid and Ghodasara, 2021).

### 4.3. Implementation Details

To keep the number of parameters as small as possible, a ResNet-18 was selected for the experiments. Further experiments with a ResNet-50 can be found in the appendix (see chapter A). The teacher and the student are each trained for 64 epochs. In the validation, those weights of the student and teacher network were used where the AUROC metric is highest. Such thresholds were then applied to the independent test set. Stochastic Gradient Descent (SGD) with a learning rate of 0.1 is used for the teacher. For the student, the learning rate is 0.5. The batch size is 2 for teachers and students. The training parameters for f-AnoGAN are the same as in Simarro et al. (2020).

## 5. Results

To evaluate performance, we followed common standards and used the area under the reciver-operator curve (AUROC) and Average Precision (AP). In addition, intersection over union and dice-coefficient were calculated to assess segments detected at pixel level.

| Exp. | Arch. | Configuration | | | Detection | | Segmentation | | |
|---|---|---|---|---|---|---|---|---|---|
| | | Patch Learning | ImageNet Weights | k-means | AUROC [%] | AP [%] | AUROC [%] | IOU [%] | DICE [%] |
| | f-AnoGAN | | | | 86.89 | 86.38 | 82.81 | 8.29 | 14.98 |
| 1 | 3D STFPM | | x | | 62.81 | 60.37 | 45.83 | 9.15 | 0.0 |
| 2 | 3D STFPM | x | | | 61.42 | 68.71 | 71.43 | 6.52 | 12.14 |
| 3 | 3D STFPM | x | x | | **94.13** | **94.38** | **89.22** | 12.09 | 20.98 |
| 4 | 3D STFPM | x | x | x | - | - | 59.72 | **17.69** | **28.17** |

Table 1: Results of the experiments on the BraTS data set. 3D STFPM with a ResNet-18 as a backbone and with pre-trained ImageNet weights, using ACS convolution, and 3D self-supervised patch learning beats f-AnoGAN.

**Metrics**    Table 1 shows the results compared to the f-AnoGAN. It should be noted that the method with ACS convolution, ImageNet initialization and 3D self-supervised patch learning achieves the best result. Both on the image level and on the pixel level. In addition, one can see that with each additional piece of information that is entered into the training (ImageNet weights, patches for the teacher, and k-Means Clustering), the performance of the independent test data set increases.

In the original paper from Wang et al. (2021), the teacher is only initialized with ImageNet weights. One can clearly see that 3D patch learning provides domain-specific knowledge. In addition, the ImageNet-weight information improves the result again. However, it should also be noted that the evaluation of the intersection over union delivers a poor result. This is because many false positives are produced and no supervised method is used for correction.

**Parameters**    The f-AnoGAN with its 3D convolutional layers requires a few more parameters during training. Generator, discriminator and encoder come to $55,029,542$ parameters. Therefore, the network can only process $(64 \times 64 \times 64)$ voxel images. In contrast, the Student Teacher approach requires only 22,346,752 parameters (11,173,376 each).

**Training-Convergence**    Through early stopping, the optimal AUROC value for detection was achieved after 12 epochs.

### 5.1. 3D Self-Supervised Patch Learning

Through self-supervised patch learning, the teacher learns to replicate the input images exactly in their intermediate layers. The network can thus represent an input image spatially well in the feature space. Pseudo-RGB images can be created by dimension reduction over the multiplied feature embedding layers, to visualize that the Euclidean distances between the individual patches are maintained in the embedding space (Danon et al., 2018). Looking at the teacher's summed output layers in figure 2 after self-supervised patch learning, one can also see that the network is quite good at tracking healthy and diseased tissue. The pseudo-RGB image, which uses dimension reduction via PCA, also suggests that Euclidean distances are preserved (Danon et al., 2018).

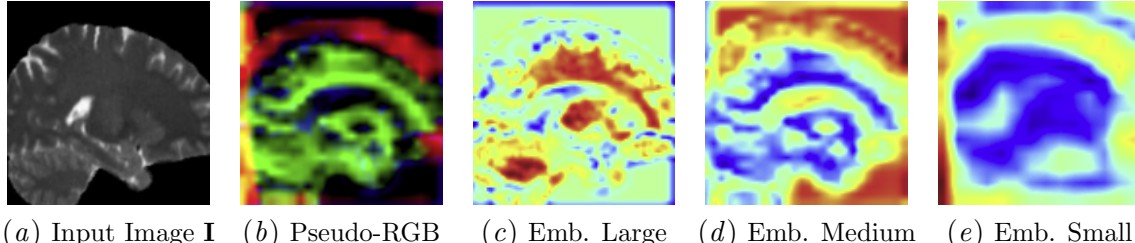

$(a)$ Input Image **I**  $(b)$ Pseudo-RGB  $(c)$ Emb. Large  $(d)$ Emb. Medium  $(e)$ Emb. Small

Figure 2: Euclidean distances are preserved both in the pseudo-RGB image and in the feature embedding layers after each ResNet block.Marked here with Large, Medium and Small for the 1st, 2nd and 3rd ResNet block respectively.

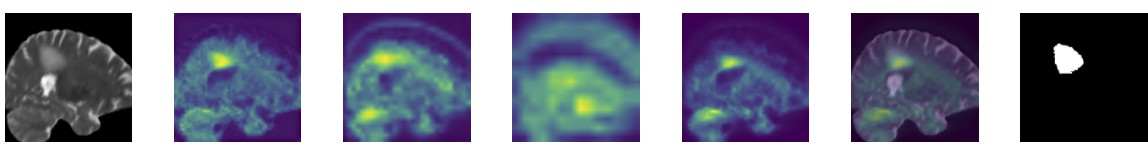

Figure 3: From left to right: Input image **I** as 2D slice from sagittal orientation of a BraTS T2 scan, anomaly map after ResNet block 1, anomaly map after ResNet block 2, anomaly map after ResNet block 3, generated anomaly map $\mathbf{A}_{map}$, anomaly map $\mathbf{A}_{map}$ placed on input image **I**, ground truth of the BraTS sequence.

### 5.2. Anomaly Maps

After the first three ResNet blocks, an anomaly map can be created by multiplying the teacher and the student. The figure 3 shows these color-encoded maps belonging to the input image on the left, as well as the intermediate anomaly maps on the different scales. The color-encoding is used to indicate the relative level of anomaly, where blue areas encode low differences and yellow areas encode the highest differences found. The BraTS dataset contains ground truth labels, which encode where abnormal tissue is in the image. When comparing the resulting anomaly map from the network to the ground truth map, one finds that these are similar, suggesting that the network can detect pathological brain formations.

### 6. Conclusion

We presented a framework for anomaly detection of 3D MRI scans that uses axial-coronal-sagittal convolution to use ImageNet pretrained networks and can simultaneously process a 3D input. With the newly introduced 3D self-supervised patch learning for the teacher, a broad knowledge of healthy and diseased tissue is taught. Together with the student's 3D training, state-of-the-art performance is achieved. Large to medium-sized lesions can be well identified thanks to the top-down approach that can process high-resolution MRI scans. In addition, the framework presented scores with short training times and can therefore be used flexibly.

## Acknowledgments

This research was funded in part by the humAIne project, funded by the German Ministry of Science and Education (FKZ 02L19C203). We would like to thank Lena Will, radiologist at the University Hospital Knappschaftskrankenhaus Bochum, for the enriching discussion on the subject of 3D MRI data.
We thank Sven Kreienbrock and Tobias Erm for technical assistance.

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

## Appendix A. Supplementary Material

**3D images** : Anomaly image after applying our 3D patch-based student-teacher framework (see figure 4).

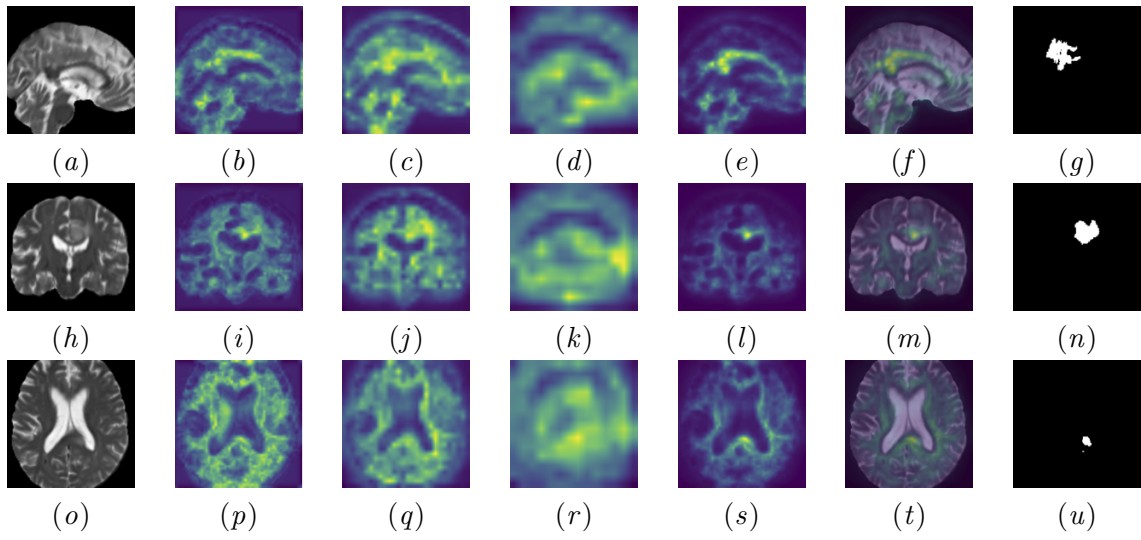

Figure 4: From top to bottom: Sagittal, -coronal, -axial alignment. With each input image **I** as a 2D slice of a BraTS T2 scan, anomaly map after ResNet block 1, anomaly map after ResNet block 2, anomaly map after ResNet block 3, generated anomaly map $\mathbf{A}_{map}$, anomaly map $\mathbf{A}_{map}$ placed on input image **I**, ground truth of the BraTS sequence.

**ResNet-Versions** : We tested different ResNet versions using ImageNet weights and 3D SSPL without k-means clustering. Results are listed in the table 2. In each case, the same learning rates were set for the teacher and for the student (see chapter 4.3).

| Configuration | Detection | | Segmentation | | |
|---|---|---|---|---|---|
| ResNet Version | AUROC [%] | AVGPREC [%] | AUROC [%] | IOU [%] | DICE [%] |
| 50 | 66.90 | 61.08 | 81.95 | 7.73 | 13.95 |
| 34 | 63.61 | 67.03 | **91.92** | **16.37** | **26.92** |
| 18 | **94.24** | **94.68** | 90.08 | **12.22** | **21.13** |

Table 2: Experiments on the BraTS dataset with ImageNet pre-trained weights and 3D self-supervised patch learning. The ResNet-18 and ResNet-34 performs best, but ResNet-18 can be train faster.

**System and runtime** : All training was conducted on a computer with an Nvidia RTX 4090 24 Gbyte graphics card, an Intel Core i7-6800K CPU, and 64 Gbytes of memory.

Student-Teacher runtime: 7 minutes per epoch are required to train the teacher. The student is trained in 3 minutes per epoch. For both networks, training is performed over 64 epochs.

f-AnoGAN runtime: On the same system, with the same data loader, the f-AnoGAN requires 3 minutes per epoch. The encoder is also trained in 3 minutes per epoch. As suggested by (Simarro et al., 2020), more than 200 epochs have been trained.

