# OpenReview forum: "A Patch-based Student-Teacher Pyramid Matching Approach to Anomaly Detection in 3D Magnetic Resonance Imaging"
_MIDL.io/2024/Conference — MIDL 2024 Poster_

### Official Review · Reviewer_VPzy · 2024-02-28

**Confidence:** 5
**Preliminary Rating:** 3
**Final Rating:** 3.5

**Summary:**

The authors propose an anomaly detection framework which uses student-teacher feature pyramid matching for detecting anomalies at image and voxel levels. They integrated self-supervised pre-training with 3D patches and axial-coronal-sagittal convolutions. They evaluate their method on BRATS and IXI.

**Strengths:**

- Interesting top-down approach for anomaly detection, especially, employment of self-supervised 3d Patch-pretraining and axial-coronal-sagittal convolutions for using the 2D ImageNet weights for 3D convolutions
- Code was made public, given reproducibility
- Implementation details given
- Clear experiments planning

**Weaknesses:**

- Only f-AnoGan for comparison, more models would have been more convincing
- An ablation study on the ACS convolution would have been interesting, in comparison to 3D convolutions and slice-wise 2D convolutions
- Brain MIR images are less variant, in comparison with, e.g., Chest or Fully-Body MRIs - another dataset with higher structural variance would have been more convincing regarding the generalisability of the proposed method

**Detailed Comments:**

- Paper needs minor writing revision, e.g. "self supervised"
- This sentence needs further explanation: "Training-Convergence We have found that after just 12 epochs, the student has trained
enough to achieve an optimal AUROC value. With this, good results can be created with little training time." - How? Early stopping or empirically?

**Justification Of Final Rating:**

The reviewer thanks the authors for their responsiveness to feedback. With the revisions addressing the reviewer's comments, the manuscript has demonstrated improvement for a 'borderline accept.' For future submissions, the reviewer recommends diversifying evaluations beyond brain datasets and with more benchmarking models to enhance the methods' applicability.

**Justification Of The Preliminary Rating:**

It is an interesting approach and seems to perform well in comparison with the 3D-conv-based f-AnoGan. An extension with more benchmark models and other structural more variant data would be more convincing.

**Questions To Address In The Rebuttal:**

Were there any preliminary comparisons made with 2D f-AnoGan and a slice-wise version of the proposed method? What would be the expectations of the authors regarding performance and training/inference time?

Were there any experiments made with other data besides Brain MRI datasets?

**Special Issue:**

No

---

> ### Author Response · Authors · 2024-03-17
>
> - Were there any preliminary comparisons made with 2D f-AnoGan and a slice-wise version of the proposed method? What would be the expectations of the authors regarding performance and training/inference time?
> 	- Thank you for the question. In fact, the first experiments were created on 2D slices from the BratS 2021 tumour dataset and the IXI dataset.
> 	- Details: For each MRI, a 2D slice was cut out of the axial plane where the tumour is largest (compare with segmentation map). For the non-lesional MRIs from the IXI dataset, the registered MRIs were cut out from the same location.
> Network architecture was similar to that in ([Wang et al. (2021)](https://arxiv.org/abs/2103.04257)).  The teacher was initialised only with ImageNet weights and pre-trained with self-supervised patch learning in 2D, as in ([Danon et al. (2018)](https://arxiv.org/abs/1807.03130)).
> Results (Classification):
> ImageNet (AUROC; AP) - 82.0; 85.1
> SSPL (AUROC; AP): 82.1; 90.0
> 	- Evaluation:
> Only one slice per MRI was used for training, validation, and testing (where the tumour occupies the largest area according to the segmentation map).
> Therefore, the results are good, but not directly comparable to the results in the paper. If you use all slices from an MRI, you can expect significantly worse results because firstly, the spatial information is lost here and secondly, you have to conduct the detection for each slice in the MRI.
> Going back to the question of whether this can reduce the training and inference time, we would instead say that it significantly increases it. Extract each slice, send it through the teacher and student, form an anomaly map, and then put all the slices back together to form a large anomaly map.
> 	- No experiments were conducted for f-AnoGAN on 2D slices.
> - Were there any experiments carried out with other data besides Brain MRI datasets?
> 	- We attempted [ATLAS](https://fcon_1000.projects.nitrc.org/indi/retro/atlas.html). The detection performance was AUROC 58.2%. No experiments were conducted for segmentation performance because the detection performance does not perform well.
> There may be room for further experiments in future work.
> - This sentence needs further explanation: "Training-Convergence We have found that after just 12 epochs, the student has trained enough to achieve an optimal AUROC value. With this, good results can be created with little training time." - How? Early stopping or empirically?
> 	- Thanks for the comment. We have specified the sentence in the paper. Early stopping was used here.

---

### Official Review · Reviewer_mNPe · 2024-02-28

**Confidence:** 5
**Preliminary Rating:** 3
**Final Rating:** 3.5

**Summary:**

In this paper, the authors propose to use a student-teacher feature pyramid matching technique (STFPM), previously developped for industrial anomaly detection, and use it, along with two other previoulsy developped techniques (Axial Coronal Sagittal convolutions) and self-supervised patch learning, to detect "anomalies", i.e. brain tumors on the BraTS dataset (public). Healthy training data is the IXI dataset (public). 4 experiments are designed to prove the benefits of some of the techniques added, and a comparison with a state-of-the-art method is done.

**Strengths:**

1) To the best of the reviewer knowledge the STFPM method was never applied to medical images
2) The databases used are public
3) Comparison is done with a SOTA method
4) Combination of STFPM + ACS + SSPL seems novel

**Weaknesses:**

1) The reviewer believes that overall the paper sometimes lacks precise descriptions of the method/evaluation. While these flaws (listed in the categories bellow) are in themselves minor, the reviewer believe that they add up to a point where the quality of the overall paper is hindered.
2) The anomalies detected, i.e. tumors from the BraTS datasets, are quite large and oftentimes very obvious to detect. While this flaw of evaluation is verry common in the current medical imaging UAD literature, the reviewer believes it still should be noted.
3) The novelty is limited, as the 3 methods used and combined here already exist (the reviewer believes that this point in itself would not imply rejection but must still be noted).

**Detailed Comments:**

- "3D magnet resonance images"  --> "magnetic"
- "to regard the recognition of disease patterns and the identification of deviations from the norm" : and --> as
- "a loss is formed over them." : please use a more precise vocabulary
- "ℓ2 distance between ℓ2 normalized feature vectors" : very minor but the reviewer believes this can be proven to be equivalent to applying cosine similarity.
- "Flair sequence" --> "FLAIR sequence"
- "To keep the parameters as small as possible"  --> number of parameters
- "learn rate" --> "learning rate"
- "The color-encoding is used to indicate the relative level of anomaly, where blue areas encode low differences and red areas", the reviewer believes the authors meant "yellow" instead of "red", judging from fig 3
- The reviewer believes the authors could cite earlier work than Bergmann et al 2019 for student teacher, such as Hinton et al Knowledge distillation

**Justification Of Final Rating:**

As described in the comments the reviewer was a little bit disappointed by a small lack of clarity and didactic in the responses but thinks this could still be presented at the conference.
...........

**Justification Of The Preliminary Rating:**

The reviewer believes the paper, if significantly improved by adressing all of the many comments above, could be accepted (weak accept), without significant clarifications and modifications though, I would reject this paper (weak reject).

**Questions To Address In The Rebuttal:**

- "this patch size turned out to be better." : could the authors precise in what way this smaller patch sized proved better ?
- "Before the patches p, p+ , p− are entered into the teacher one after the other, the patches are scaled to the spatial size of the input images I"  : I did not understand this sentence, are the patches in the end of the same size as the whole image ? If this is the case how is the interpolation done ? Does it make sense to provide a Resnet trained with whole image a patch largely upsampled ?
- F̂sl (I) is here a normalization : what kind of normalization ? Please precise.
- Could the authors precise the scale factor between the three maps of the pyramid ?
- "Experiment 4: Same as Experiment 3, but k-means clustering is applied in the evaluation of the Amap anomaly map, as suggested by Siddiquee et al. (2019)" : the reviewer did not find in the provided reference any mention of k-means. A small sentence precising how is k-means applied and on what would be appreciated.
- "For comparability, all images are registered on the template MNI-152" : is it possible to precise how this registration step is necessary for comparability ? It seems that the developped method could be applied without registration.
- "To generate a binary segmentation map, all values greater than 0 are considered a lesion." : can the authors precise what the labels mean when refering to the BraTS dataset ?
- "For the evaluation, those weights of the student were chosen where the AUROC score is highest" :  can the authors precise this sentence ? AUROC on detection or segmentation ? Is this on the validation set ? If it is on the test set this overestimates the performances.
- The reviewer believes the details on how the detection was done were not provided : the proposed method gives a per-voxel anomaly score map, then how is this score map transformed into a unique score for whole-image detection ?
- The reviewer did not understand why the detection performances were not reported for exp 4
- "Generator, discriminator and encoder come to 55, 029, 542 parameters. Therefore, the network can only process (64 × 64 × 64) voxel images."  : why ? on what laptop/computer configuration ?
- On all the presented visual exemples, from the 3 pyramid maps, the "grosser" systematically missed the tumor, can the authors comment on what this gross-scale map brings to the method ?
- "figure 2 after self-supervised patch learning, one can also see that the network is quite good at tracking healthy and diseased tissue." : can you please precise what the reader is suppose to see ?
- "which uses dimension reduction via PCA, also suggests that Euclidean distances are preserved" : can the authors provide additional details about this experiments ?
- As additional comparison, the reviewer believes the authors could add performances from the literature, i.e. raw AUROC/etc. provided by the papers doing UAD on BraTS, in addition to the re-implementation of f-anogan made by the authors.
- "the framework presented scores with short training times and can therefore be used flexibly." : can the authors provide the training time ? and a comparison with f-anogan ?

**Special Issue:**

No

---

> ### Author Response · Authors · 2024-03-17
>
> - Q 1:
> 	- 3D boxes of the size ($32 \times 32 \times 32$) voxels are cut out for the teacher's self-supervised image patch learning. In contrast to the 2D approach from [1], which suggests a patch size of ($16 \times 16$), we found that on 3D MRIs larger boxes lead to better validation metrics. In the paper from [1], the patch size is also introduced as an unexplained hyperparameter.
> - Q 2:
> 	- Thank you for this important comment. Indeed, this step is unusual, which is why we decided to run all experiments 2 to 4 without upscaling the patches in pretraining (basically, it was only one line of code that needed to be changed).
> The results are almost identical. Therefore, we also regenerated all images in the paper for consistency. Thank you for taking a closer look.
> - Q 3:
> 	- This refers to the $ \ell_2 $ normalisation of the features [2]. We made it more precise in the paper.
> - Q 4:
> 	- The feature maps are output according to ResNet blocks 1, 2 and 3 and combined into an anomaly map. The resolutions are (in (c, x, y, z) format: (64, 56, 56, 56), (128, 28, 28, 28), (256, 14, 14, 14), respectively.
> - Q 5:
> 	- While the Fixed-Point-GAN paper by [3] mentions the k-Means step only marginally, the clustering is of quite crucial importance. As thorough inspection of the source code unveils ([Fixed-Point-GAN](https://github.com/mahfuzmohammad/Fixed-Point-GAN/blob/master/solver.py)), k-Means is used for the test results in order to eliminate the many false positives. We followed the same approach for the same purpose.
> - Q 6:
> 	- As the IXI and BraTS data each come from different sources, the established standard pipeline of registration, skullskripping, standardisation and normalisation was used.
> Registration was primarily used to remove the skullcap from the IXI data using the MNI template mask.
> - Q 7:
> 	- The tumor was divided into different regions in the BraTS dataset: "Annotations comprise the GD-enhancing tumor (ET - label 4), the peritumoral edema (ED - label 2), and the necrotic and non-enhancing tumor core (NCR/NET - label 1)" (see [5]).
> - Q 8:
> 	- We present AUROC for both detection and segmentation (see table 1). We used thresholds obtained from the validation set and applied them to the independent test set (terminology following [Ripley textbook](https://www.stats.ox.ac.uk/~ripley/PRbook/ )), which constitutes the most careful approach to evaluate performance.
> 	- The AUROC value for detection is used here for early stopping.
> 	- We have now specified this in the paper thanks to your comment.
> -  Q 9:
> 	- Image level evaluation: The largest value from the feature embeddings (i.e. anomaly map) is used for this. This value is entered into the evaluation function (AUROC and AP) together with the ground truth.
> 	- Pixel level evaluation: Here, the feature embeddings are transferred to the evaluation function (AUROC, DICE, IOU) together with the ground truth map.
> 	- We have now specified this in the paper thanks to your comment.
> - Q 10:
> 	- In experiment 4, k-Means was applied based on the knowledge that there is only one contiguous tumor spot per MRI [3]. For the detection performance, whether an MRI is lesional or non-lesional, k-Means has no advantages.
> - Q 11:
> 	- We have now added the information in the paper (see appendix).
> - Q 12:
> 	- We followed the practice from [2]. This may be of interest to investigate in future work. In our present work, this is not possible due to space constraints.
> - Q 13:
>  	- This is an error, it should read Figure 3, not Figure 2.
> - Q 14:
> 	- Figure 2 basically shows that the feature embeddings also look similar to an MRI input image in the intermediate layers. PCA is used to convert the higher dimensional embeddings into a pseudo RGB image. We have followed Danon et al. [1].
> -  Q 15
> 	-  The same data set is not always used as a basis. In the example of [4], it is not specified exactly which version of the BraTS data set is used as the basis. The [website (braintumorsegmentation)](http://www.braintumorsegmentation.org/ ) states that the versions differ greatly from one another.
> - Q 16:
> 	- Training student-teacher:
> 		- Teacher: 7 minutes per epoch
> 		- Student: 3 minutes per epoch
> 		- For both networks, the training is carried out over 64 epochs.
> 	- Training f-AnoGAN:
> 		- Generator-Discriminator: 3 minutes per epoch.
> 		- Encoder: 3 minutes per epoch.
> 		- Just as in the f-AnoGAN paper [6], over 200 epochs were trained.
> 	- We made it more precise in the paper.
>
> References:
> - [1] [Danon et al. (2018)](https://arxiv.org/abs/1807.03130)
> - [2] [Wang et al. (2021)](https://arxiv.org/abs/2103.04257)
> - [3] [Siddiquee et al (2019)](https://arxiv.org/abs/1908.06965)
> - [4] [Luo et al. (2023)](https://www.sciencedirect.com/science/article/pii/S0010482523000756)
> - [5] [Baid et al. (2021)](https://arxiv.org/pdf/2107.02314.pdf)
> - [6] [Simarro et al. (2020)](https://arxiv.org/abs/2010.04717)

---

> > ### Comment · Reviewer_mNPe · 2024-03-24
> >
> > The reviewer thanks the authors for the clarifications and answers. The lack of highlighted difference with the former PDF (e.g. modifications in red) and the numbering of questions instead of quoting made the answer a little bit difficult to read. Some parts of the comments were addressed and modified while others remain, which is a little pitiful (e.g. keeping "values greater than 0 are considered a lesion" instead of precising edema, tumor core, etc.). For many comments the authors did not quite explain their reasoning but rather just referred to a reference (e.g. "We followed the practice from [2]"), which the reviewer thinks is a little bit of a shame because it would have been of interest to expose the authors choices and reasoning. Overall many comments still address the questions and modifications have been made accordingly.

---

> > > ### Author Response · Authors · 2024-03-26
> > >
> > > Thank you for your feedback, we highly appreciated the broad and detailed comments and questions. All major changes have now been highlighted in red. We have made our best possible efforts to explicitly phrase our own choices and reasoning within the character limit for the authors' response. Whenever our choices and reasoning followed previously published arguments, we covered our approach by citing the corresponding reference. In some cases, particularly in the involvement of k-means in the fixed-point GAN, the choices are somewhat hidden in the details of previous work.

---

> > > > ### Comment · Reviewer_mNPe · 2024-03-26
> > > >
> > > > The reviewer thanks the authors for the clarifications made.

---

### Official Review · Reviewer_Kx2D · 2024-02-28

**Confidence:** 3
**Preliminary Rating:** 4
**Recommendation:** Poster
**Final Rating:** 4

**Summary:**

The paper is interested in, especially small and subtle, anomaly detection in 3D MRI images. It leverages a Student Teacher Feature Matching that operate comparisons at 3 pyramid levels to form an anomaly map. The teacher is trained with both pathological and healthy patients using self-supervised patch learning extended to 3D, and leveraging ACS convolutions to initialize the model with ImageNet weights. The student is trained only with healthy patients. The method is evaluated on brain tumor segmentation using data from the BraTS tumor dataset and IXI dataset for healthy patient data. It compares favorably with respect to state of the art f-anoGAN approach with excellent detection performance and promising segmentation results.

**Strengths:**

The paper is clearly written and well structured. Furthermore, it is evaluated on public datasets, and the authors' code is available online. The performance scores are excellent in detection, considering it is a (almost) unsupervised approach. The experiments are based on a comparison with an approach from the state of the art, that is different in spirit, and reports on various configurations of the method to outline the impact of each contribution.

**Weaknesses:**

- The method is a smart combination of previously published methods, with contributions to adapt them to 3D images.
- One could expect that more reference methods are included in the experiments, for comparison. In particular, training the proposed model is (very, but still) weakly supervised since the teacher model must be trained with both pathological and healthy patients and the student with only healthy patients. But f-AnoGAN is unsupervised. The performance assessment would be more convincing if a weakly supervised method was included (see e.g. https://arxiv.org/abs/2302.04549 associated with https://github.com/yzhao062/WSAD to take your pick)
- the authors claim their interest in small and subtle anomaly detection, but their method is only applied to brain tumor segmentation. It would have been interesting to have performance assessment on a second clinical question, with smaller anomalies, to support their claim.
- The k-means step, even though it improves the segmentation results, seems like an ad-hoc post-processing step to polish things up. How does it behave if a healthy patient is tested?
- Numerical details are missing concerning data splits for training (section 4.2). How was f-anoGAN trained?

**Detailed Comments:**

- in the abstract: magnet -> magnetic
- ACS convolution kernel sizes in section 3.1, equation 1 should be KxKx1, Kx1xK and 1xKxK
- first line of page 8: euclidean -> Euclidean
- first line of section 5.2: by multiply the output -> multiplying

**Justification Of Final Rating:**

The authors answered all my questions, and added details to the paper that improve its readability. I appreciate and thank the authors for their efforts. Despite its clear interest to MIDL attendance, I am not keen on raising my rating to strong accept, which should be reserved for papers that are highly innovative in deep learning, in my opinion.
Please be aware that the paper now slightly exceeds the 8-page limit.

**Justification Of The Preliminary Rating:**

The paper is well written and structured, and describe a smart combination of techniques, with contributions to adapt them to 3D images. The experiments are reproducible, given that some more details are needed concerning data splits, and f-anoGAN training.

**Questions To Address In The Rebuttal:**

- details on data splits for training models
- the number of false positives seems to be an issue. Could the authors set some more insights on the false positive rate?
- comment on the fairness of the experiments, considering that the proposed model is not purely unsupervised

**Special Issue:**

No

---

> ### Author Response · Authors · 2024-03-17
>
> - details on data splits for training models:
> 	- The T2 sequences of the BraTS and IXI data were each divided into 70\% training data, 15\% & validation data and 15\% test data.
> 	- We made it more precise in the paper.
> - the number of false positives seems to be an issue. Could the authors set some more insights on the false positive rate?
>  	- Indeed, as we mention, “many false positives are produced and no supervised method is used for correction”. This comment refers to false activation within MRIs that contain anomalies, rather than false positive identification of non-lesional MRIs. This can also be observed in the f-AnoGAN output ([Simarro et al.](https://arxiv.org/abs/2010.04717)), but is not discussed by the authors. The annotated brain tumors in the datasets may coincide with other anomalies that are not annotated as tumors, which would require detailed discussion.
> - comment on the fairness of the experiments, considering that the proposed model is not purely unsupervised:
> 	- We highly appreciate raising this question. There is indeed a difference between the prior knowledge required by our approach compared to prior knowledge required in the f-AnoGAN. The f-AnoGAN requires MRIs that are known to be non-lesional for training. Our approach, on the other hand, trains the teacher model on lesional and non-lesional MRIs (without knowing what is what), while only the student requires training on data known to be non-lesional.
> - How was f-anoGAN trained?
> 	- All parameters were used as suggested by [Simarro et al. (2020)](https://arxiv.org/abs/2010.04717). We have now specified this in the paper.
>
> *Comment:*
> - Thank you for the suggestions. Details about the splits and training details for f-AnoGAN have been added to Section 4.2. Thanks to your comment, it was explicitly pointed out in the paper that k-means is only applied to the lesional MRIs (segmentation task).

---

### Meta-Review · Area_Chair_JFYp · 2024-04-02

**Recommendation:** Accept (Poster)
**Confidence:** 4

**Metareview:**

All reviewers found the proposed method to be novel and the results promising. 1 weak accept and 2 borderline accept recommendations from reviewers.

---

### Decision · Program_Chairs · 2024-04-05

Accept (Poster)